# Increasing Endoglin Deletion in Endothelial Cells Exacerbates the Severity of Brain Arteriovenous Malformation in Mouse

**DOI:** 10.3390/biomedicines12081691

**Published:** 2024-07-30

**Authors:** Zahra Shabani, Leandro Barbosa Do Prado, Rui Zhang, Wan Zhu, Sonali S. Shaligram, Alka Yadav, Calvin Wang, Hua Su

**Affiliations:** 1Center for Cerebrovascular Research, University of California, San Francisco, CA 94143, USA; zahra.shabaninabikandi@ucsf.edu (Z.S.); leandro.barbosa87@gmail.com (L.B.D.P.); zrnjcn@gmail.com (R.Z.); ms.wanzhu@gmeil.com (W.Z.); sonali.shaligram@gmail.com (S.S.S.); alka.yadav@ucsf.edu (A.Y.); calvin.wang@ucsf.edu (C.W.); 2Department of Anesthesia and Perioperative Care, University of California, San Francisco, CA 94143, USA

**Keywords:** arteriovenous malformations, endothelial cells, endoglin, hereditary hemorrhagic telangiectasia

## Abstract

Endoglin (*ENG*) mutation causes type 1 hereditary hemorrhagic telangiectasia (HHT1). HHT1 patients have arteriovenous malformations (AVMs) in multiple organs, including the brain. In mice, *Eng* deletion induced by R26RCreER or SM22αCre leads to AVM development in the brain and other organs. We hypothesized that an increase in Eng- negative ECs will enhance AVM severity. To increase EC *Eng* deletion, we used a codon-improved cre (icre), which is more potent in recombination of the floxed alleles than the wild-type (WT) cre. R26RCreER;*Eng*^f/f^ mice that have a Rosa promoter driving and tamoxifen (TM)-inducible WT cre expression globally, and PdgfbiCreER;*Eng*^f/f^ mice that have a Pdgfb promoter driving and TM-inducible icre expression in ECs were treated with three intra-peritoneal injections of TM (2.5 mg/25 g of body weight) to delete *Eng* globally or in the ECs. AAV-VEGF was stereotactically injected into the brain to induce brain focal angiogenesis and brain AVM. We found that icre caused more *Eng* deletion in the brain, indicated by a lower level of Eng proteins (*p* < 0.001) and fewer Eng-positive ECs (*p* = 0.01) than mice with WT cre. Mice with icre-mediated *Eng* deletion have more abnormal vessels (*p* = 0.02), CD68^+^ macrophages (*p* = 0.002), and hemorrhage (*p* = 0.04) and less vascular pericyte and smooth muscle coverage than mice with WT cre. In addition, arteriovenous shunts were detected in the intestines of icre mice, a phenotype that has not been detected in WT cre mice before. RNA-seq analysis showed that 8 out of the 10 top upregulated pathways identified by gene ontology (GO) analysis are related to inflammation. Therefore, the increase in *Eng* deletion in ECs exacerbates AVM severity, which is associated with enhanced inflammation. Strategies that can reduce Eng-negative ECs could be used to develop new therapies to reduce AVM severity for HHT1 patients.

## 1. Introduction

Hemorrhagic telangiectasia (HHT) is an autosomal-dominant disorder occurring in about 1 in 5000 people worldwide [1]. HHT is characterized by vascular lesions in multiple organs, such as arteriovenous malformations (AVMs) in the brain and visceral organs, and telangiectasias (small AVMs) in mucocutaneous [2]. AVMs are a complex of abnormal vessels, connecting arteries and veins directly without normal intervening capillary beds [3].

More than 80% of HHT patients have mutations in endoglin (*ENG*, HHT1) or activin receptor-like kinase 1 (*ALK1* also known as ACVRL1, HHT2) genes [4]. Ninety percent of HHT patients show recurrent and spontaneous epistaxis (nosebleeds) [5,6]. Intracranial hemorrhage (ICH), caused by the rupture of brain AVMs, is the most devastating and life-threatening symptom [7,8]. All available managements for HHT have limited efficacy or have considerable adverse effects. There is no FDA-approved treatment for HHT.

There are multiple mouse models that recapitulate HHT AVM and telangiectasia created through a global or endothelial cell (ECs)-specific knockout of *Eng*, *Alk1*, or *Smad4* during the embryonic stage, neonatal period, or adulthood [1,9,10,11,12]. Studies using HHT mouse models and surgically resected brain AVM specimens uncovered that the angiogenesis and mutation of AVM causative genes in ECs are key factors for AVM initiation in an *ENG* or *ALK1* mutant subject [13,14], although this may not apply to the AVM development in the retina of *Smad4* EC-deleted mice [15]. Inflammation also plays an important role in AVM progression and hemorrhage [16,17].

One of the interesting findings is that the mutation of the *Alk1* gene in a small portion of ECs or bone marrow-derived ECs is enough to trigger brain AVM formation in the presence of angiogenic stimulation [18]. The overexpression of *Alk1* in all ECs by using a transgenic technique rescued a phenotype in *Alk1* and *Eng* null mice, without any untoward effect [14]. It will not be easy to introduce the *ALK1* gene to all human ECs. Since the number of *Alk1*-negative ECs is positively correlated with the AVM severity [18], the introduction of the ALK1 gene to a fraction of ECs of HHT2 patients will be able to reduce AVM severity. However, it is unclear if the AVM severity in HHT1 is also correlated with the number of ENG-negative ECs.

To address this question, we used a PdgfbiCreER transgene that expresses a codon-improved cre (icre) [19] to enhance *Eng* deletion in ECs. We found that icre mediated more reduction in Eng expression in brain AVMs, which is associated with more severe brain AVM phenotypes and inflammation.

## 2. Materials and Methods

### 2.1. Ethics Statement

All animal experimental protocols were approved by the Institutional Animal Care and Use Committee (IACUC) of the University of California, San Francisco (UCSF). The staff in the IACUC of UCSF Animal Core Facility provided animal husbandry according to the guidance of certified Animal Technologists. Veterinary care was offered by the IACUC faculty and veterinary inhabitants located on the San Francisco General Hospital campus. All mice were maintained in a pathogen-free area in 421 × 316 cm^2^ cages and were kept on a 12 h light and dark cycle with free access to food and water. Animal experiments were conducted by certified authors who contributed to the study.

### 2.2. Animals

Three groups of 8- to 10-week-old mice in C57BL/6 backgrounds were used: (1) wild type (WT); (2) R26RCreER;*Eng*^f/f^ mice that have a Rosa promoter driving tamoxifen (TM)-inducible WT cre expression and have *Eng* gene exons 5 and 6 flanked by loxp sites [20]; and (3) PdgfbicreER;*Eng*^f/f^ mice that have a Pdgfb promoter driving TM-inducible iCre (codon improved cre) [19] expression and have *Eng* gene exons 5 and 6 flanked by loxp sites. An equal number of male and female mice were used.

### 2.3. AVM Model Induction

AVMs in *Pdgfb*icreER;*Eng*^f/f^ and R26RCreER;*Eng*^f/f^ mice were induced through intra-brain injection of an adeno-associated vector expressing vascular endothelial growth factor [AAV-VEGF, 2 × 10^9^ viral genomes (vgs)] on day 1 to induce brain angiogenesis that is needed for the induction of brain AVM and intra-preoperational (i.p.) injection of TM (2.5 mg/kg of mouse body weight) for three consecutive days, starting on the day of intra-brain injection of AAV-VEGF to delete *Eng* gene [8]. Brain samples and intestines were collected 8 weeks after model induction (Figure 1).

For the intra-brain injection of viral vectors, the mice were anesthetized with 4% isoflurane inhalation and were placed in a stereotactic apparatus with a mouth holder (David Kopf Instruments, Tujunga, CA, USA). A burr hole was drilled in the pericranium 2 mm lateral and 1 mm posterior to the bregma. A 10 µL Hamilton syringe was inserted into the right basal ganglia 3 mm beneath the brain surface. Two microliters of viral suspension were slowly injected at a rate of 0.2 µL per minute. The needle was withdrawn after 10 min, and the wound was closed with a 4–0 suture [8].

### 2.4. Western Blot

Mouse brains were collected after anesthetizing the mice with 4% isoflurane inhalation. Brain tissues (1 mm^3^) containing the vector injection sites were collected. Protein was extracted from brain tissues using a cell lysis buffer (Cell Signaling, Danvers, MA, USA) supplemented with 1 mM PMSF (Cell Signaling) and quantified by the Bradford method (Bio-Rad, Hercules, CA, USA) or the BCA method (Thermo Scientific, Emeryville, CA, USA) using a microplate reader (Emax, Molecular Devices, Sunnyvale, CA, USA). Protein samples were loaded and run into 4–20% Tris-Glycine gels (Bio-Rad, Hercules, CA, USA) and transferred onto PVDF membranes (Bio-Rad). Immunoblotting was performed using primary antibodies specific to Eng (1:200, R&D Systems, Minneapolis, MN, USA) and Gapdh (1:1000, Abcam, Cambridge, UK). A goat anti-rabbit IgG antibody and a donkey anti-goat antibody (Li-Cor, Lincoln, NE, USA) were used as the secondary antibodies. Eng and Gapdh bands were detected by a Li-Cor Quantitative Western blot scanner and quantified using Li-Cor imaging software (Li-Cor).

### 2.5. Immunofluorescence Staining

After being anesthetized with isoflurane inhalation, Cy5-fluorescein-conjugated lycopersicon esculentum lectin (Vector Laboratories, Burlingame, CA, USA) was injected via the jugular vein of one set of the mice to stain ECs. These mice were then perfused with heparinized PBS 1 h later through the left cardiac ventricle to clear blood from vessels, followed by 4% paraformaldehyde. The brains of these mice were collected and incubated in 10% Neutral Buffered Formalin containing 20% sucrose for 2 days, and then frozen in dry ice. Another set of brains was collected from anesthetized mice and frozen in dry ice directly. Brains were sectioned into 20 μm thick sections using a Leica CM1950 Cryostat (Leica Microsystems, Wetzlar, Germany).

Two sections per brain adjacent to the injection site were selected and incubated at 4 °C overnight with the following primary antibodies: rat anti-CD31 antibody (1:100, Cat #SC-18916, Santa Cruz Biotechnology, Santa Cruz, CA, USA) or goat anti-mouse CD31 antibody (1:250, Cat #AF3628, R&D Systems) to stain the ECs, goat anti-mouse Eng antibody (1:100, Cat #AF1320, R&D Systems) to detect Eng expression, rabbit anti-mouse α smooth muscle actin (α SMA) antibody (1:400, Cat #A2547, Sigma, St Louis, MO, USA) to stain vascular smooth muscle cells, and rat anti-mouse CD68 antibody (1:250, MCA1957, Bio-Rad) to stain activated microglia and macrophages.

A donkey anti-rat antibody conjugated with Alexa Fluor 488-conjugated (1:100, Cat #A-21208), a donkey anti-goat antibody conjugated with Alexa Fluor 594 (1:300, Cat #A-11058), a donkey anti-rabbit antibody conjugated with Alexa Fluor 555 (1:400, Cat #A-31572), and a donkey anti-rat antibody conjugated with Alexa Fluor 594 (1:400, Cat #A-21209, Thermo Fisher Scientific, Waltham, MA, USA) were used as the secondary antibodies to visualize positive stains.

After being incubated with the secondary antibodies, all sections were mounted with Vectashield antifade DAPI containing mounting medium (Cat #H-1200, Vector Laboratories, Burlingame, CA, USA), examined, and imaged using a Keyence fluorescence microscopy under a 20× objective lens (Model BZ-9000, Keyence Corporation of America, Itasca, IL, USA). A total of six images were taken from each brain sample, three from each brain section (to the right, to the left, and below the injection site).

### 2.6. Latex Perfusion

Mice were deeply anesthetized using isoflurane inhalation. The abdominal and thoracic cavities were opened. Both left and right atria were cut off. Blue latex dye (1 mL, Connecticut Valley Biological Supply Co., Southampton, MA, USA) was injected into the left cardiac ventricle using a 27-gauge needle attached to a 5 mL syringe. The brains and intestines were harvested and fixed with 10% Neutral Buffered Formalin overnight. The intestines were imaged directly. The brains were dehydrated with methanol series and clarified with benzyl alcohol/benzyl benzoate (1:1 ratio). After clarification, the brains were cut coronally into 3 mm thick slices directly using a razor blade and imaged.

### 2.7. Prussian Blue Staining

The brains were collected from anesthetized mice and frozen in dry ice directly. Brains were sectioned into 20 μm thick sections using a Leica CM1950 Cryostat (Leica Microsystems, Wetzlar, Germany). Two sections per brain adjacent to the injection site were used for detecting iron deposition using an Iron Stain Kit (Sigma-Aldrich, St. Louis, MO, USA). Slides were incubated in a freshly prepared working iron stain solution for 15 min, washed in distilled water, and then counterstained with pararosaniline solution for 3 min. Data are presented as the percentage of the Prussian blue-positive area versus the total hemisphere area.

### 2.8. RNA Sequencing

Brain tissues (1 mm^3^) around AAV-VEGF injection sites and corresponding areas on the contralateral sides were collected. The total RNAs were isolated from these tissues and sent to Novogene Co (Sacramento, CA, USA) for sequencing using the company’s standard protocol (Appendix A). The outcome data were analyzed by Novogene Co.

### 2.9. Statistical Analysis

All quantifications were performed by at least two researchers who were blinded to the treatment groups. The images were coded by a researcher who did not participate in the quantification. Data were analyzed using GraphPad Prism 8.

Sample sizes were calculated based on priori sample size analysis with the following assumptions: α = 0.05, β = 0.2 (power 80%). For example, for vSMAs, the power calculation indicated that n = 5 allows us to detect the difference between the WT cre and icre group with 80% power. The significant results suggest that the powers for all the analyses were sufficient, whereas insignificant results might have been attributed to a lack of difference between WT cre and icre groups.

Western blot and immunofluorescence staining were performed to detect Eng expression in WT, cre, and icre groups. We used 6 mice per group, and comparisons among multiple groups were performed by one-way ANOVA, followed by Tukey’s post hoc test. To analyze the vascular density (WT cre: n = 6 and icre: n = 9), dysplasia index (WT cre: n = 6 and icre: n = 8), expression of pericytes (WT cre: n = 6 and icre: n = 5), vSMCs (WT cre: n = 6 and icre: n = 8), and macrophages (WT cre: n = 5 and icre: n = 5), un-paired *t*-test was used for two sample comparisons. Due to a non-normalized distribution, data of microhemorrhages (WT cre: n = 6 and icre: n = 6) were analyzed by a nonparametric Mann–Whitney test. Data are presented as mean ± standard deviation (SD). A *p*-value of ≤0.05 was considered significant. Sample sizes are indicated in figure legends.

## 3. Results

### 3.1. Codon-Improved Cre (Icre) Is More Effective than WT Cre in the Deletion of Eng in Brain ECs

Due to the location of *Eng*^f/f^ in the chromosome, it is difficult to be recombined by WT cre. The injection of 2.5 mg/25 g of body weight has been identified as the most effective regimen for R26RCreER;*Eng*^f/f^ mice [21]. To study if the number of Eng negative ECs is correlated with AVM phenotype severity, we tested if icre [19] can delete the *Eng* gene from more ECs than WT cre. We treated R26RCreER;*Eng*^f/f^ mice that have WT cre and *Pdgfb*iCreER;*Eng*^f/f^ mice that have icre with the same dose of TM (2.5 mg/25 g) for 3 consecutive days at the initiation of model induction. We found that, compared with WT cre mice, icre mice have a lower level of Eng proteins in their brains (WT cre versus icre: 44.3 ± 15.1% of the mean of wild-type mice versus 7.1 ± 4.4%, *p* < 0.001, Figure 2A,B). Immunofluorescence staining showed fewer Eng-positive ECs in the brain AVMs of icre mice (5.8 ± 2.0% of total ECs) than WT cre mice (25.8 ± 9.3%, *p* < 0.01) and the brain angiogenic region of WT mice (79.5 ± 15.3%, *p* < 0.001, Figure 2C,D). These data indicate that icre is more effective in recombining *Eng* floxed alleles than WT cre.

### 3.2. Enhanced Eng Deletion in ECs Decreased Pericyte and Vascular Smooth Muscle Coverage in Brain AVM

We found that icre mice have more abnormal vessels (vessels with lumen size > 15 μm, icre vs. WT cre: 11.0 ± 3.2/mm^2^ vs. 6.75 ± 2.31/mm^2^, *p* = 0.02), less vascular pericyte coverage (icre vs. WT cre: 62.8 ± 7.79% of CD13-positive cells/CD31-positive cells vs. 75.33 ± 4.17%, *p* = 0.007), and fewer vascular smooth muscle positive vessels than WT cre mice (icre vs. WT cre: 0.34 ± 0.15% of SMA^+^ vessels/total vessels that have lumen size > 15 μm vs. 0.67 ± 0.15%, *p* = 0.001). Vascular densities were not significantly different between the two groups (Figure 3).

The number of Eng-negative ECs was positively correlated with the number of abnormal vessels in brain AVMs (r = 0.97; Figure 3F). Collectively, these data suggest that the higher number of Eng-negative ECs is associated with more severe phenotypes in Eng-deficient mouse brain AVMs.

### 3.3. Increased Eng Deletion in ECs Enhanced CD68^+^ Microglia/Macrophage Infiltration and Hemorrhage in Brain AVMs

To test whether increased Eng-negative ECs enhance inflammation in brain AVMs. We quantified activated microglia and macrophages on CD68 (a marker for activated microglia and macrophages) and CD31 antibody-stained sections. We found more CD68^+^ cells in the brain AVMs of icre mice ((58.0 ± 26.18/mm^2^) than WT cre mice (87.1 ± 8.89/mm^2^, *p* = 0.002, Figure 4A,B).

An increase in Eng-negative ECs in icre mice also increased microhemorrhages in brain AVMs (0.45 ± 0.48% of Prussian blue^+^ area/total hemisphere area) compared with the brain AVM of the WT cre mice (0.006 ± 0.004%, *p* = 0.04, Figure 4C,D). Therefore, an increase in the number of Eng-negative ECs enhances inflammation and hemorrhage in brain AVMs.

### 3.4. Arteriovenous (AV) Shunts Developed in the Intestines of Icre Mice

Previous studies did not detect any AV shut in R26RCre;*Eng*^f/f^ mice that have WT cre [21].

To determine whether the lack of AV shunt in the intestine of WT cre mice is due to insufficient *Eng* deletion in ECs, we casted the vessels with latex dye. The particles in latex dye are too large to pass the capillary. After intra-cardiac left ventricle infusion, the dye enters the vein only when there is an AV shunt. The AV shunts were detected in the intestines of 20% of icre mice. Consistent with previous data, no AV shunt was detected in the intestines of WT cre mice (Figure 5).

### 3.5. Increase in Eng Deletion in ECs Upregulated Pro-Inflammatory Pathways in Brain AVMs

To identify changes in pathways and the expression of genes in the brain AVM of icre mice compared with the brain AVM in WT cre mice, RNA-seq was performed. There were 1810 genes differentially expressed between the brain AVM tissues of icre and WT Cre mice; 946 (52.2%) were upregulated and 864 (47.7%) genes were downregulated in the brain AVMs of icre mice. To identify biological functional pathways, we used gene ontology (GO) analysis and found that, compared with WT cre mice, 8 of the top 10 upregulated biological pathways in icre mice are related to inflammation and 2 are related to angiogenesis and vascular genesis (Figure 6A). No top 10 downregulated pathways are related to vascular biology in the brain AVMs of icre mice versus the WT cre group (Appendix A). *Kyoto Encyclopedia of Genes and Genomes* (KEGG) analysis identified the downregulation of vascular smooth muscle contraction (*p* = 0.002) in icre mice.

The top differentially expressed genes between the icre group and the WT cre group are presented in Figure 6B. The genes related to angiogenesis, inflammation, and leukocyte migration among the top 100 upregulated genes are listed in Appendix A.

## 4. Discussion

In this study, we evaluated the relationship between the number of Eng-negative ECs and AVM severity. We showed that the number of Eng-negative ECs is positively correlated with the severity of mouse AVM.

The correlation of the number of Alk1-negative ECs with brain AVM severity has been demonstrated by us previously through using different doses of TM [18]. Due to the location of *Eng* floxed alleles in the chromosome, they are difficult to be recombined by cre [22]. The injection of 2.5 mg/25 g of body weight TM has been identified as the most effective regimen for R26RCreER;*Eng*^f/f^ mice [21]. An increase in TM dose may not increase *Eng* deletion in R26RCreER;*Eng*^f/f^ mice. In this study, we used codon-improved cre (icre) [19] to increase *Eng* deletion. We also used a Pdgfb promoter to drive icre expression in ECs. We showed that icre is more potent than WT cre in the deletion of the *Eng* gene in ECs.

We then analyzed the influence of Eng-negative ECs on AVM severity. We found that, like Alk1, the number of Eng-negative ECs is positively correlated with AVM severity. Compared with WT cre mice, icre mice have more abnormal vessels and less vascular pericyte and smooth muscle cell coverage in brain AVMs. Brain AVMs of icre mice also have more activated microglia and macrophages and more severe hemorrhage than WT cre mice. RNA-seq analysis revealed that the enhanced brain AVM severity is mostly associated with the upregulation of pro-inflammatory pathways. Although 2 pathways among the top 10 upregulated pathways in icre mice are associated with angiogenesis and vascular genesis, we did not detect a significant increase in vessel density in the brain AVMs of icre mice compared with WT cre mice, which could be due to the large lumen abnormal vessels occupying more area than capillaries. These data are consistent with our prior studies that pro-inflammatory and innate immune signaling are upregulated in *Eng* knockout mice compared with wild-type mice [23], which may facilitate leukocyte infiltration and extraction.

In this study, we showed that more activated microglia/macrophages accumulated in the brain AVM of icre mice than WT cre mice. It has been demonstrated that *ENG* deficiency leads to EC hyper-permeability through the destabilization of the endothelial barrier function and the reduction in vascular mural cell coverage [24,25]. The knockout of *Eng* in mouse ECs upregulates pro-inflammatory and innate immune signaling [23]. In addition, prior studies reported an increased macrophage burden in brain AVMs with AVM causative gene mutation deleted in ECs [15]. Therefore, the increase in activated microglia/macrophages in the brain AVM of icre mice could be due to the increase in inflammatory cytokine release from inflamed Eng-negative ECs, leakage of blood content, and hemorrhage resulting from blood–brain barrier (BBB) damage. We have also noticed persistent infiltration and delayed clearance of activated microglia/macrophages in the brain angiogenic of *Eng*-deficient mice [26]. Therefore, increased *Eng* deletion in ECs could increase brain AVM severity through enhancing inflammation.

It has been shown that Eng deficiency impairs monocyte homing to the injury site [27,28,29]. Decreased homing is linked to the inability of monocytes to respond to stromal cell-derived factor 1α (SDF-1α). The fewer activated microglia/macrophages in the brain AVMs of WT cre mice could be due to *Eng* deletion in the monocyte because the R26R promoter mediates cre expression in all cells. However, we found in our previous studies that systemic *Eng* deletion causes temporal differences in macrophage responses to injuries. *Eng*^+/−^ mice showed fewer CD68^+^ cells in the peri-infarct area at 3 days but more at 60 days after stroke injury in mouse [30]. After angiogenic stimulation, the *Eng*-deleted mice had fewer CD68^+^ cells at 2 weeks and more at 8 weeks in the brain angiogenic region, compared with WT mice [26]. In the present study, the brain AVM samples were collected 8 weeks after model induction from both icre and WT cre mice. Therefore, fewer activated microglia/macrophages in the brain AVMs of WT cre mice are unlikely caused by Eng deficiency in monocytes.

Interestingly, AV shunts were detected in 20% of icre mice, but not in WT cre mice. We and some others thought that the deletion of *Eng* in mice could not induce AV shunt in the intestine [21]. Our finding in this study indicates that the previous conclusion was incorrect. The reason why no AV shunt was detected in WT cre mice is that the deletion of *Eng* gene in ECs was inadequate.

## 5. Conclusions

Our data indicate that AVM severity is correlated with the dose of *Eng*-negative ECs. It has been shown that the overexpression of Alk1 using a transgene in *Alk1* null or *Eng* null mice rescues AVM phenotypes [14]. However, it is difficult to restore ALK1 or ENG expression in all ECs in HHT patients. Our findings in this paper suggest that any strategies that can restore Eng expression in some ECs will be able to reduce disease severity in HHT1 patients.

## Figures and Tables

**Figure 1 biomedicines-12-01691-f001:**
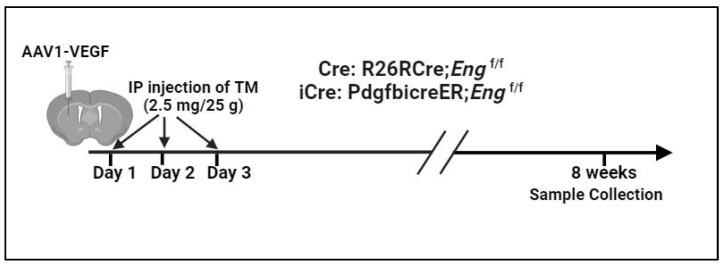
Induction of AVM models. AAV1-VEGF: AAV-VEGF vector packaged in AAV serotype 1 capsid. IP: intra-peritoneal injection; Cre: WT cre; icre: codon-improved cre.

**Figure 2 biomedicines-12-01691-f002:**
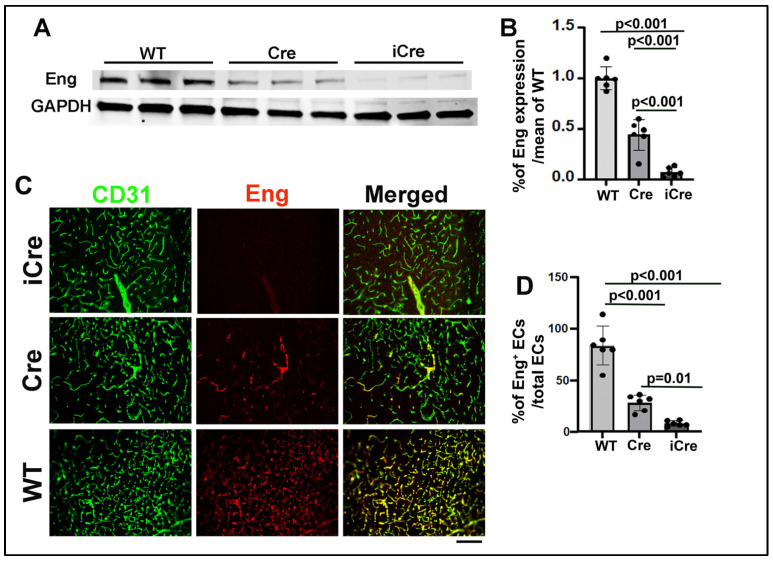
Codon-improved icre-induced *Eng* deletion in more brain ECs than WT cre. (**A**). Representative Western blot images. (**B**). Quantification of Eng expression. (**C**). Representative images of brain sections. ECs were stained by an anti-CD31 antibody (green), and Eng protein was stained by an anti-Eng antibody (red). Scale bar: 50 μm. (**D**). Quantification of Eng-positive ECs. WT: wild-type mice; Cre: R26RCreER;*Eng*^f/f^ mice; and iCre: *Pdgfb*iCre;*Eng*^f/f^ mice. n = 6.

**Figure 3 biomedicines-12-01691-f003:**
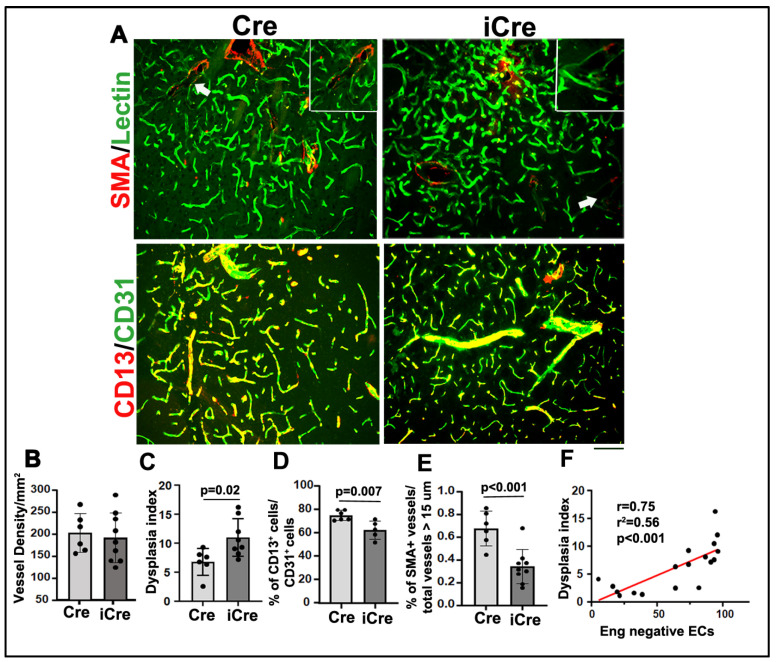
Brain AVM phenotype is more severe in icre mice than WT cre mice. (**A**). Representative images of brain sections. ECs were stained by intravascular perfusion of fluorescent labeled lectin (green). Vascular smooth muscles and pericytes were stained by anti-αSMA and anti CD13 antibodies (red), respectively. Arrows indicate dilated dysplastic vessels showing in the inserts on the upper-right corners of the pictures. Scale bar: 50 μm. Quantification of vessel density (**B**), Dysplasia Index (number of vessels with lumen size > 15 μm/mm^2^) (**C**), % of CD13-positive cells/CD31-positive cells (**D**), and % of SMA^+^ vessels/total vessels > 15 μm (**E**). WT: wild-type mice; Cre: R26RCre;*Eng*^f/f^ mice; and iCre: *Pdgfb*iCre;*Eng*^f/f^ mice. n = 5–8. (**F**) Correlation of Eng-negative ECs and dysplasia index.

**Figure 4 biomedicines-12-01691-f004:**
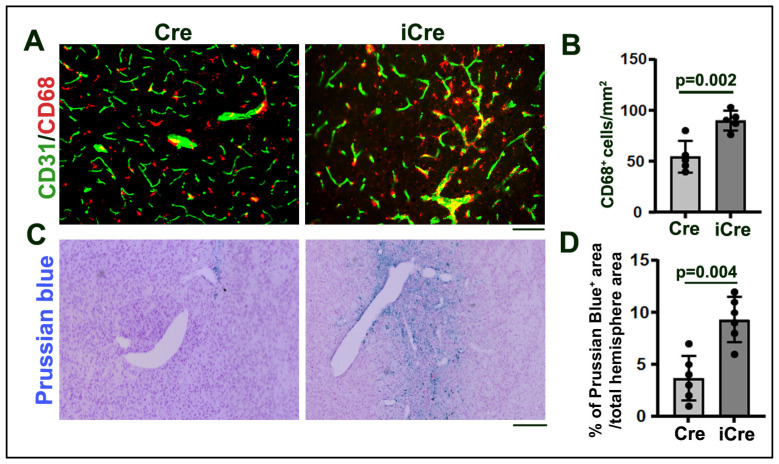
Increase in Eng-negative ECs enhanced microglia and macrophage infiltration and microhemorrhage in brain AVMs. (**A**). Representative images. ECs were stained by an anti-CD31 antibody (green) and activated microglia, and macrophages were stained by an anti-CD86 antibody (red). Scale bar: 50 μm. (**B**). Quantification of CD68^+^ cells. (**C**). Representative pictures of Prussian blue stained sections. Iron depositions at the microhemorrhage region were stained blue. Scale bar: 50 μm. (**D**). Quantification of the percentage of Prussian blue-positive area in the total area of the hemisphere. Cre: R26RCre;*Eng*^f/f^ mice; iCre: *Pdgfb*iCre;*Eng*^f/f^ mice. n = 5.

**Figure 5 biomedicines-12-01691-f005:**
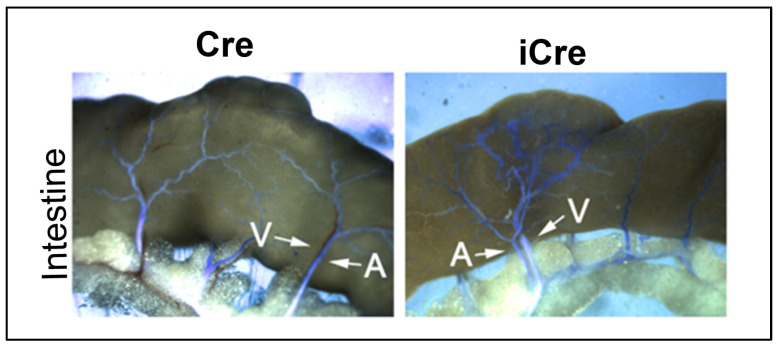
Arteriovenous (AV) shunts developed in the intestines of icre mice. Latex dye (blue) entered some of intestinal veins of icre mice. No latex-positive vein was detected in the WT cre mice. Arrows indicate arteries (A) and veins (V) in the intestines. Scale bar: 1 mm. Cre: R26RCre;*Eng*^f/f^ mice; iCre: *Pdgfb*iCre;*Eng*^f/f^ mice.

**Figure 6 biomedicines-12-01691-f006:**
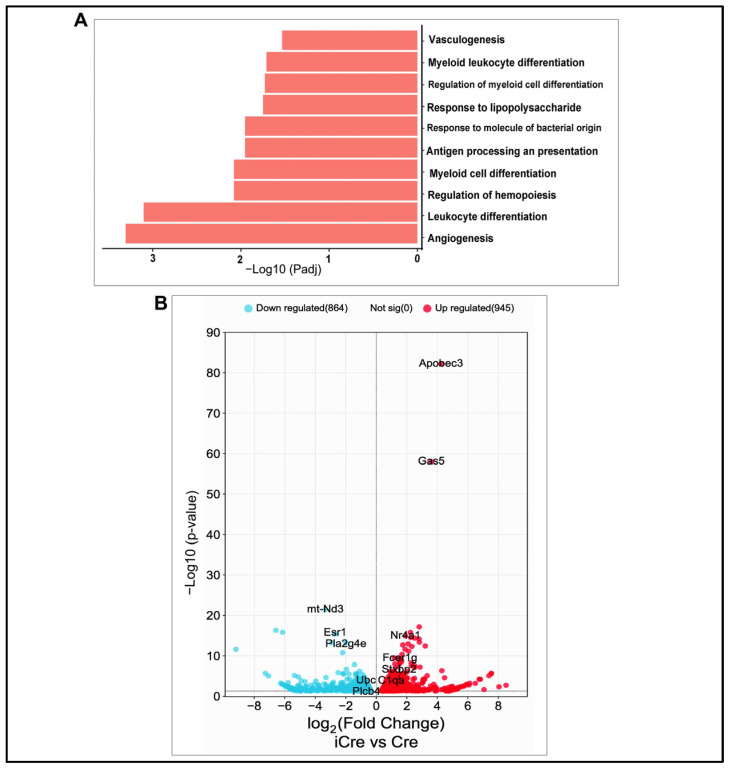
Upregulated biology pathways and differentially expressed genes. (**A**) Top 10 upregulated biology pathways. (**B**) Volcano plot. Red indicates significantly upregulated genes, whereas blue indicates significantly downregulated genes in the brain AVMs of the icre group compared with the WT cre group.

## Data Availability

The authors declare that all data supporting the findings of this study are available in the paper and its Appendix A.

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
