# Peer review of "Increasing Endoglin Deletion in Endothelial Cells Exacerbates the Severity of Brain Arteriovenous Malformation in Mouse"

_biomedicines, 2024, doi:10.3390/biomedicines12081691_

Round 1

Reviewer 1 Report

Comments and Suggestions for Authors

Overview of the manuscript

The work focuses on analysing increased in the expression of Endothelial cells (EC) Eng negative in the brain, as cause of increased arteriovenous malformations (AVM) disease. The authors explore their hypothesis on an experimental animal model of mice in which a differential deletion in the expression of the Eng gene has been performed. A codon improved cre (icre), more potent in recombination of floxed alleles than the wild type (WT) cre, was used. Transgenic mice with a promoter driving tamoxifen (TM) inducible for  cre or icre expression was used. The authors find that mice with iCre - Eng deletion have more abnormal vessels, macrophages, hemorrhage, and less vascular pericyte and smooth muscle coverage than mice with WT cre. RNA analysis showed upregulation of gene related to inflammation. The authors conclude that the increase of Eng deletion in ECs exacerbates AVM severity, which is associated with enhanced inflammation.

GENERAL COMMENT

The work is interesting and give further continuation to a previously covered topic, adding further information on the physiological pathways of AVM. The experimental transgenic mice model is an interesting and innovative model promoting similar use of transgenic model. The experimental plan is well performed, and the results provide adequate support for the discussion and conclusions.

SPECIFIC COMMENTS

Abstract

Pag. 1 line 13: the deletion of Eng is cause of HHT1. Correct the expression.

Pag. 1 line 15-18: in the abstract the report of previous work is useless, it remains confusing. Delete the paragraph.

Materials and Methods

Pag. 4, line 149-163:  Latex perfusion and Prussian blue staining, were the experiments performed on paraffin embedded sections? Explain better.

Results

Pag. 5 line 190: a repetition occurs. Correct it.

Author Response

We thank this reviewer for his/her positive comments and great suggestions. We have addressed your comments and revised the manuscript based on our best understanding of the questions.

1- Regarding the Abstract Section  

Pag. 1 line 13: the deletion of Eng is cause of HHT1. Correct the expression.

Response: Thank you for identifying this error. We have corrected the expression to “Endoglin (ENG) mutation causes type 1 hereditary hemorrhagic telangiectasia (HHT1)”.

Pag. 1 line 15-18: in the abstract the report of previous work is useless, it remains confusing. Delete the paragraph.

Response: Thank you for this suggestion. we have deleted the report of previous work.

1- Regarding the Materials and Methods

Pag. 4, line 149-163:  Latex perfusion and Prussian blue staining, were the experiments performed on paraffin embedded sections? Explain better.

Response: For latex perfusion, after clarifying with benzyl alcohol/benzyl benzoate, the brains were cut coronally into 3 mm thick slices using a razor blade. For Prussian blue staining, fresh frozen brain sections (20 µm-thick) were used.

We have added explanations in the corresponding sections.

2- Regarding the Materials and Methods

Pag. 5 line 190: a repetition occurs. Correct it.

3- Regarding the results

Response: Thank you for identifying this error. “Mice” was repeated twice. We corrected it.

Reviewer 2 Report

Comments and Suggestions for Authors

This is a very important research that demonstrates how the increase of endoglin deletion in endothelial cells increases the severity of ATV malformations in mouse.

In general, the manuscript is well written and the design of the study was sound and the experiments were well performed. In my opinion, the weakness of this study is related to the statistical analysis. In this sense, it is necessary that the authors describe the number of animals used per group and how the statistical power of the study was calculated. This information should be included in the manuscript. 

Author Response

We thank this reviewer for his/her positive comments and great suggestions. We have addressed the your comments and revised the manuscript based on our best understanding of the questions.

Reviewer comment

In general, the manuscript is well written and the design of the study was sound and the experiments were well performed. In my opinion, the weakness of this study is related to the statistical analysis. In this sense, it is necessary that the authors describe the number of animals used per group and how the statistical power of the study was calculated. This information should be included in the manuscript. 

Response: We thank you for raising these comments and providing suggestions. Sample sizes were calculated based on priori sample size analysis with the following assumptions: α=0.05, β=0.2 (power 80%). For example, for vSMAs, the power calculation indicated that n=5 allows us to detect the difference between the WT cre and icre group with 80% power. The significant results suggest that the powers for all the analyses were sufficient, whereas insignificant results might have been attributed to a lack of difference between WT cre and icre groups.
